# Promising Strategies for the Development of Advanced In Vitro Models with High Predictive Power in Ischaemic Stroke Research

**DOI:** 10.3390/ijms23137140

**Published:** 2022-06-27

**Authors:** Elise Van Breedam, Peter Ponsaerts

**Affiliations:** Laboratory of Experimental Hematology, Vaccine and Infectious Disease Institute (Vaxinfectio), University of Antwerp, 2610 Antwerp, Belgium; peter.ponsaerts@uantwerpen.be

**Keywords:** stroke, in vitro models, ischaemia, brain spheroids, brain organoids, microfluidics

## Abstract

Although stroke is one of the world’s leading causes of death and disability, and more than a thousand candidate neuroprotective drugs have been proposed based on extensive in vitro and animal-based research, an effective neuroprotective/restorative therapy for ischaemic stroke patients is still missing. In particular, the high attrition rate of neuroprotective compounds in clinical studies should make us question the ability of in vitro models currently used for ischaemic stroke research to recapitulate human ischaemic responses with sufficient fidelity. The ischaemic stroke field would greatly benefit from the implementation of more complex in vitro models with improved physiological relevance, next to traditional in vitro and in vivo models in preclinical studies, to more accurately predict clinical outcomes. In this review, we discuss current in vitro models used in ischaemic stroke research and describe the main factors determining the predictive value of in vitro models for modelling human ischaemic stroke. In light of this, human-based 3D models consisting of multiple cell types, either with or without the use of microfluidics technology, may better recapitulate human ischaemic responses and possess the potential to bridge the translational gap between animal-based in vitro and in vivo models, and human patients in clinical trials.

## 1. Introduction

Stroke is one of the leading causes of death and disability worldwide [1]. In the majority of cases (~62%), stroke is caused by occlusion of an arterial vessel by an embolus or thrombus, referred to as ischaemic stroke [2]. The interruption of blood supply to the brain depletes the brain tissue from oxygen and other nutrients, causing energy failure and triggers the activation of a cascade of events eventually leading to brain damage [3]. Processes of this ischaemic cascade include excitotoxicity, oxidative stress, blood–brain barrier (BBB) dysfunction, inflammation and cell death of neurons, glia and endothelial cells [4,5]. These events can result in ischaemic necrosis within minutes for the brain tissue exposed to the most drastic blood flow reduction. This irreversibly damaged brain tissue is known as the ischaemic core. Surrounding the ischaemic core is the ischaemic penumbra, which contains cells that are less severely affected and that are potentially salvageable from a lethal fate. However, without improved perfusion or therapeutic intervention to improve the resistance of cells to injury, the ischaemic cascade occurring in the penumbra will result in secondary cerebral damage thereby expanding the infarct core several hours to days after the stroke onset [4,5,6,7].

Notwithstanding the impact of a stroke on the patient’s quality of life and on society, the current treatment of ischaemic stroke patients is limited to the administration of the thrombolytic agent tissue plasminogen activator or to mechanical clot retrieval by thrombectomy. However, only a small proportion of all acute ischaemic stroke patients are eligible for the last-mentioned treatments, mainly due to the very narrow therapeutic time window after stroke onset [8]. Tremendous efforts have been made to find new therapies targeting the ischaemic cascade to prevent injured or vulnerable neurons in the ischaemic penumbra from dying or even to stimulate regenerative processes. Over a thousand candidate neuroprotective drugs have been proposed, showing promising results in animal models. Unfortunately, none of those led to an effective therapy to date as many of the neuroprotective agents failed when translated to the clinic. Multiple reasons may account for this lack of success, such as deficiencies in animal studies or the clinical trial design [9,10], but it is equally clear that the predictive power of the systems currently used to model the ischaemic stroke in vitro and as such to validate candidate compounds should be questioned. 

In this review, we first describe the general experimental set-up to model ischaemic stroke in vitro, including the current main cellular platforms. Next, we describe the main factors affecting the predictive power of in vitro models, thereby shedding light on in vitro ischaemic stroke research for the future.

## 2. Modelling Ischaemic Stroke In Vitro

Most of the knowledge on the pathophysiological mechanisms of an ischaemic stroke is derived from animal-based in vitro and in vivo models. Over the past decades, different animal models of stroke have been developed, induced by emboli, intraluminal suture, photothrombosis or endothelin-1, typically in rodents [3,7,11]. The rat is one of the most commonly used species in stroke research, among other reasons, due to the similarity of the cerebral vasculature and physiology with that of humans. Moreover, mice are often used, since they are helpful in unravelling the function of certain genes in the pathophysiology of stroke by means of the creation of transgenic mice [3,7,11]. Animal stroke models have been an indispensable tool, as they can model different aspects of the complex pathophysiology of ischaemic stroke that cannot be modelled (yet) in simple in vitro models lacking intact blood vessels and blood flow [3,12]. However, simplified, highly controlled in vitro systems are required and preferred when investigating specific basic mechanisms and cell type-specific responses under ischaemia-like conditions [7,12]. Besides, in the context of testing potential neuroprotective compounds, working in vitro allows high-throughput screenings, even on a human-based background [12].

### 2.1. Inducing Ischaemia-like Conditions In Vitro

In vitro models of ischaemic stroke typically mimic the conditions of the ischaemic penumbra—the target tissue for therapeutic intervention—where cells are functionally silent but initially viable. To study ischaemic stroke in vitro, ischaemia-like conditions can be achieved by different approaches. The most common and most physiologically relevant way to induce ischaemia-like conditions is by so-called ‘oxygen-glucose deprivation’ or OGD. In this approach, cell or tissue cultures are placed in a hypoxic or anaerobic chamber, containing a N_2_/CO_2_ atmosphere, where the O_2_/CO_2_ equilibrated medium becomes replaced by the glucose-free N_2_/CO_2_ equilibrated medium at the start of incubation [13,14,15,16,17,18,19]. The cultures are maintained for a duration of 30 min up to 24 h in the chamber, depending on the specific cell type used and the desired degree of ischaemic damage. Typically, a longer duration of oxygen and glucose deprivation is needed to cause cell injury or death in vitro than in vivo. Compared to ischaemia in vivo, adenosine triphosphate (ATP) depletion is less severe and the release of glutamate is delayed [7]. OGD is often terminated by glucose addition and reoxygenation and is cultured under ‘normal’ conditions for up to 24 h prior to downstream analyses. This allows modelling of in vivo reperfusion, known to further aggravate ischaemic injury [20].

Besides OGD, hypoxia can be induced through either chemical or enzymatic inhibition of cellular metabolism. The chemical method relies on inhibition of the mitochondrial electron transport chain and has been regularly applied to cell cultures to study ischaemic stroke. For instance, sodium azide and antimycin are commonly used chemical-hypoxia inducers in these studies [21,22,23,24]. Less common is the enzymatic induction of hypoxia, which relies on manipulating the glucose oxidase and catalase (GOX/CAT) system [25,26,27]. Though less physiologically relevant, these chemical and enzymatic approaches can result in hypoxic/ischaemic injury in a shorter time frame than conventional OGD [28].

Due to implementation of novel technologies in in vitro stroke model development, recently, researchers were able to recapitulate another factor besides oxygen and glucose depletion, namely the interrupted blood flow, by employing microfluidic systems [23,29]. This appears to be another factor affecting the downstream ischaemic cascade by reducing the integrity of the BBB and thereby allowing it to mimic in vivo stroke even more closely.

Moreover, specific aspects of the ischaemic cascade can be modelled. For example, excitotoxicity models have been developed by exposing cultures to glutamate or glutamate receptor agonists such as N-methyl-D-aspartate (NMDA) [30]. The increase in the levels of intracellular free calcium is also an important effector of secondary injury subsequent to an ischaemic insult and has been simulated in in vitro models by thapsigargin treatment [31].

### 2.2. Most Common Cellular Platforms in In Vitro Stroke Research

The main cellular platform used for in vitro stroke research consists of monocultures of rodent primary neurons. In general, the use of monocultures is preferred when studying cell-specific responses to OGD and/or to evaluate the action of neuroprotective compounds on specific cell types. Among a lot of other applications, primary rat neurons have been used to evaluate the protective effect of the basic fibroblast growth factor [32], intermittent hypothermia [33] and oxytocin against damage induced by an ischaemic insult [18], as well as to elucidate the mechanisms underlying neuronal autophagy in ischaemic stroke [34]. Moreover, rat primary neurons have been used to study the effect of hypoxia on the neuronal activity by plating them on multi-electrode arrays during exposure of the culture to different durations of hypoxia [35].

Another widely used platform to model ischaemia-like damage are organotypic brain slice cultures, typically from rodent origin. In these cultures, brain slices are obtained from young animals (postnatal day P3 to P10) and allowed to further develop and mature in vitro [36,37]. The advantage of this culture type is that it largely preserves tissue structure maintaining neuronal activities and synapse circuitry [38]. Moreover, since multiple cell types are present, this model additionally allows one to study cell–cell interactions [38]. Due to these unique features, this system is closer to an animal model than cell culture. Organotypic brain slice cultures have been valuable in the study of pathogenic mechanisms leading to ischaemia-induced neuronal cell death, in particular with the excitotoxic mechanism. For instance, the involvement of glutamate—accumulating extracellularly after an ischaemic insult—and glutamate receptors and transporters in the excitotoxic-induced damage have been extensively studied using the brain slice model [39,40], reviewed in detail by Noraberg et al. [41]. Related or not to this glutamate-induced damaging mechanism, brain slice models have been applied to study calcium overload, mitochondrial dysfunction and oxidative stress, as well as to evaluate neuroprotective drugs [41]. Furthermore, in contrast to nearly all other in vitro systems where OGD media is applied over the entire culture, brain slice cultures could also be used as a platform to mimic focal ischaemia. A protocol by Richard et al. describes a focal ischaemia model by focally applying OGD medium to a small portion of the brain slice while bathing the remainder of the slice with normal oxygenated media [42].

Together with animal models, monocultures of primary rodent-derived neurons and rodent organotypic brain slices have shaped stroke research until the present. These platforms have increased our understanding of the ischaemic cascade and unveiled a myriad of potential targets for neuroprotective therapies. However, the high attrition rate of potential neuroprotective compounds in clinical studies should make us aware of the limitations of current models to model human ischaemic stroke with sufficient fidelity. As such, the ischaemic stroke field would greatly benefit from the implementation of novel, more complex in vitro models with improved physiological relevance next to traditional in vitro and in vivo models in preclinical studies, to more accurately predict clinical outcomes. In what follows, we will elaborate on the main factors that define the predictive value of in vitro stroke models, including the origin or source of cells or tissue, the presence of other central nervous system (CNS) cell types in co-culture models, and the dimensionality of culture and the use of advanced technologies, such as microfluidics (Figure 1).

## 3. Factors Defining the Predictive Value of In Vitro Ischaemic Stroke Models

### 3.1. Origin of Cells or Tissue Used for In Vitro Models of Ischaemic Stroke

As described above, in vivo and in vitro rodent-based models are standards used in stroke research. Their use has led to our increased understanding of the ischaemic cascade of human stroke as the main aspects of stroke hold true across all mammals. However, as rodents and humans are separated by 80 million years of evolution [10], species-specific anatomical, cellular and molecular differences exist between humans and rodents potentially affecting the outcome of neuroprotective strategies.

At the anatomical level, differences between human and rodents are evident, with humans having large gyrencephalic brains with a high proportion of white matter, whereas rodents have small smooth brains with relatively little white matter [10]. Associated to this difference in brain anatomy, the number of outer radial glia cells in rodent brains is small, while in primates this cell type is more abundant and possesses a higher self-renewal capacity [43,44]. Furthermore, species-specific differences have been reported on the expression levels and function of several BBB-transporters [45,46]. Likewise, comparison of the distributions of predominant glial glutamate transporters revealed significant differences between species [47]. This variation may translate into differences in pathophysiological stroke mechanisms or available targets between species. Specifically related to stroke, it has been demonstrated that the duration of excitotoxity after the ischaemic insult differs between mice and humans, with a longer duration for humans [48]. Moreover, at the immunological level, important differences exist between rodents and humans. A pioneering study of Seok et al. compared genomic responses to different acute inflammatory stresses (including endotoxemia, burns and trauma) between humans and mice, and found that the responses elicited in humans are not reproduced in the mouse models [49]. Moreover, there is increasing evidence that there are important differences between human and murine microglia [50]. Moreover, in the context of ischaemic stroke, dissimilarities are becoming apparent [51]. A study by Du et al. demonstrated that the baseline expression of cytokines/chemokines and response after OGD and reoxygenation in primary neurons, astrocytes and microglia differed significantly between rodents and humans [52]. For instance, while human primary neurons showed a downregulation in many of the determined chemokines (CX3CL1, CXCL12, CCL2, CCL3, and CXCL10) after OGD and reoxygenation, mouse neurons showed a mixed response with the up- and downregulation of the same chemokines. These findings exemplify the importance of using human-based in vitro models in fundamental as well as translational stroke research, next to traditional in vivo models. The introduction of human-based in vitro models in the preclinical phase of drug discovery and development would allow target identification and proof-of-principle demonstration that attacking these targets elicits appropriate cellular responses in a human context before entering the clinic, increasing chances of success for the agents to be effective in clinical setting [10]. Nevertheless, the use of human-based in vitro systems is rare in the field of ischaemic stroke. The few human-based systems that have been used to date consist mainly of transformed cell lines and primary human brain slice preparations, each associated with their own limitations.

Most of the human-based studies were performed with immortalised neuroblastoma cell lines, such as SH-SY5Y cells [53,54,55]. Though interesting when considering future high-throughput screening applications, cell lines do not always accurately replicate the physiology of primary cells. Moreover, in ischaemic stroke research, their limited physiological relevance is reflected by their reduced susceptibility to hypoxic stimuli and their constant proliferation when compared to primary neurons [10]. Similar for in vitro stroke models of the BBB or neurovascular unit (NVU), brain endothelial cell lines, such as HCMEC/D3, show lower protein expression of tight junctional proteins, adhesion molecules and transporters, as compared to their in vivo counterpart, possibly affecting the outcome of studies [28].

In contrast to cell lines, primary human brain slice preparations are highly physiologically relevant. The few studies employing human brain slices were focused on the excitotoxic component of the ischaemic cascade [56,57,58]. The major issue to use these models is the extremely limited availability to human brain tissue. Moreover, caution should be given to the interpretation of results since the brain tissue is often derived from the neurosurgery of young epileptic patients, and preparation of the slices can introduce trauma possibly confounding results [9]. Considering similar limitations, retrospective studies using the post-mortem brain tissue of human ischaemic stroke patients are extremely limited but highly valuable. The few publications existing using human post-mortem stroke tissue all belong to the same research group and report on the ischaemia-induced alterations in gene expression [59,60,61].

For decades, the limited availability and physiological relevance of human in vitro systems and the lack of technological advancements have favoured the use of rodent-based systems over human-based systems. Fortunately, human pluripotent stem cells have provided another cell source for generating human-based in vitro models with the ability to overcome the aforementioned limitations. A recent publication of Liu et al. [62] describes human embryonic stem cell (ESC)-derived neurons as an alternative model for ischaemic stroke research. Besides human ESCs, the advent of induced pluripotent stem cell (iPSC)-technology enabled pluripotent stem cells to be made out of terminally differentiated adult somatic cells, such as dermal fibroblasts and peripheral blood mononuclear cells [63,64]. Since its discovery, protocols to generate different neural cells, such as neurons, astrocytes, oligodendrocytes and microglia, but also endothelial cells have been developed [65,66,67,68,69,70,71,72,73,74,75,76,77,78]. More recently, this technology has found its way in the ischaemic stroke research. A first study using human iPSC-derived neurons was performed in 2020 by Juntunen et al., where the effect of OGD and potential protection by adipose stem cells was investigated [79]. Furthermore, human iPSC-derived cells are also increasingly being employed in the context of BBB/NVU models [23,80,81,82]. It should be noted that, though iPSC-derived in vitro platforms have boosted research in many fields and hold great promise for the future, there are still challenges associated with the use of iPSCs. Residual epigenetic memory, genetic background and incomplete reprogramming could possibly influence the iPSC phenotype and differentiation potential, resulting in a great diversity among human iPSC-derived cell lines [83]. The reproducibility may partially be increased by the improvement and standardisation of differentiation protocols with the identification of environmental cues involved in neural development in the field of developmental biology.

### 3.2. Multicellular Co-Culture Models for In Vitro Ischaemic Stroke Research

As mentioned earlier, the majority of in vitro stroke research is conducted using monocultures of neurons. Apart from neurons, a monoculture of rodent primary astrocytes has been used to determine the protective roles of pinin and stem-cell derived exosomes after ischaemic stroke [84,85]. Furthermore, when focusing on the BBB-disruption facet under ischaemic conditions, the use of pure cultures of brain endothelial cells has been regularly reported [86,87,88,89,90,91,92]. Monoculture systems are particularly useful to investigate mechanisms restricted to specific cell types or to determine the contribution of specific cell types to different pathophysiological mechanisms. However, several reasons substantiate the use of co-culture models to obtain models better resembling the human brain. First, the human brain consists of an intricate cellular network, including neurons, astrocytes, oligodendrocytes, microglia, pericytes and endothelial cells. Therefore, every cell type added to the in vitro system increases the complexity, approaching more the in vivo complexity of the human brain. Second, co-cultures enable cellular interactions that occur in vivo and as such the presence of different cell types and interactions can influence RNA transcription, protein production and functionality of certain cell types.

The importance of cell–cell interactions occurring under physiological conditions become evident from different publications. For example, astrocytes provide metabolic substrates to neurons (i.e., energy supply to neurons) and are actively involved in the formation and refinement of neuronal networks. Indeed, they are demonstrated to integrate and modulate neuronal excitability and synaptic transmission [93,94,95]. These functions of astrocytes could also be observed in in vitro astrocyte-neuron co-culture models. Astrocytes, from rodent and human origin, co-cultured with human PSC-derived neurons improves the functional maturation of those neurons, as demonstrated by an increased percentage of active neurons, bursting frequency and synchronisation of neuronal calcium oscillations when compared to the neuronal monocultures [96,97,98,99]. Moreover, mutual interactions between microglia and neurons in the healthy brain exist, where neurons (e.g., through CX3CR1-CX3CL1 or CD200-CD200R interactions), or neural environment in general, keep microglia in a non-activated state, thereby favouring their homeostatic functions maintaining neuronal health and regulating proper function of neuronal networks [70,100,101,102]. Furthermore, co-cultures of brain endothelial cells with other CNS cells, such as astrocytes and pericytes, contribute to BBB integrity and function among others by stimulating tight junction formation and expression of polarised transporters in endothelial cells [28,83,103,104,105,106].

Also under pathological ischaemic conditions, cellular interactions are important in regulating cell behaviour and contribute to the mechanisms leading to brain injury or recovery. For example, co-cultures of microglia/macrophages with neurons or brain slices have been developed and employed in the field of stroke research to investigate the inflammatory response secondary to an ischaemic insult. After an ischaemic insult, brain-resident microglia and blood-derived macrophages can acquire a pro-inflammatory neurotoxic phenotype, further exacerbating brain damage. To study the cross-talk between hypoxic neurons and macrophages, Desestret et al. subjected an organotypic hippocampal slice to OGD for 30 min and subsequently added macrophages for 2 days [107]. Other studies used co-cultures of rat primary microglia with primary neurons or a combination of primary neurons and astrocytes to elucidate the effect of neuronal ischaemia on microglia polarisation and, conversely, the effect of microglia phenotype on the fate of healthy or ischaemic neurons [13,108,109]. These studies confirmed that the pro-inflammatory activation of microglia by damage-associated molecular patterns released from damaged neurons after OGD further exacerbates neuronal death. Likewise, a neutrophil-neuronal co-culture was recently developed to investigate mechanisms of neutrophil-dependent neurotoxicity [110]. The last-mentioned study found that cell–cell contact was required for the process of neutrophil-induced neuronal injury. Next to neuro-immune interactions, neurovascular and gliovascular interactions occurring during cerebral ischaemia have also been identified. From a study comparing brain endothelial cells in monoculture versus co-cultures of brain endothelial cells with neurons or astrocytes, it became apparent that neurons and astrocytes, exposed to either OGD, aglycemia or hypoxia, affect different endothelial properties, including its barrier and lymphocyte adhesion properties, endothelial cell adhesion molecule expression and in vitro angiogenic potential [111]. For instance, the interaction of brain endothelial cells with neurons or astrocytes under OGD and subsequent reoxygenation, results in attenuation of BBB permeability and in recovery of the barrier. This compensatory mechanism of astrocytes for maintaining BBB function after ischaemic stroke has been confirmed in another study, identifying a role for astrocyte-derived pentraxin 3 [112]. However, the excessive production of cytokines, chemokines and proteases in the ischaemic infarct might undermine the adaptive nature of the BBB, leading to increased permeability [111]. Identification of these interactions is important as changes in BBB permeability can affect cerebral oedema, post ischaemic brain angiogenesis (associated with survival of stroke patients) and leukocyte interactions that aggravate ischaemia reperfused stroke brain damage [111].

All these examples represent only a small part of all existing (un)identified interactions occurring under ischaemic stroke-pathological conditions that can affect the progression of ischaemia-associated brain damage or recovery. Therefore, it is of importance to include different cell types to more faithfully recapitulate the ischaemic responses occurring in vivo, ideally in ratios representative of the adult human brain (e.g., glia/neuron ratio of less than 1:1) [113]. Besides the aforementioned co-cultures of microglia, macrophages or neutrophils with neurons, other co-cultures consisting of neurons and astrocytes have been used in ischaemic stroke research [16,114]. Recently emerging three-dimensional (3D) models of the brain also consist of multiple cell types, which will be further discussed in the next section ‘Dimensionality’. In addition, BBB/NVU models of ischaemic stroke often combine different cell types, which will be further discussed under the ‘BBB/NVU models’ section.

### 3.3. Dimensionality of Cell Culture Models for In Vitro Ischaemic Stroke Research

Most of the knowledge derived from in vitro stroke studies is based on neural cells grown as monolayers. This traditional simplified culture system has been of undisputable significance for biomedical research, including stroke, especially considering their relatively low cost and reproducibility when compared to animal models [115]. Moreover, decades of research using these monolayer cultures has led to the optimisation and standardisation of many downstream applications tailored for 2D cultures, including the easy visualisation by means of microscopic imaging. Nevertheless, 2D cultures are unable to mimic the complicated microenvironment cells experience in tissue. Unlike cells cultured in 2D, in the in vivo brain, cells are able to generate 3D projections and establish multiple interactions with other cells and cell types and the extracellular matrix (ECM) [115,116], eventually affecting their morphology, survival, proliferation, differentiation, gene expression and even function (e.g., electrophysiological network properties) [31,115]. Therefore, 3D models of the brain are considered more realistic models of the human brain than conventional 2D models, better mimicking its complexity and possibly ischaemia-induced responses [115].

A first model that allows ischaemic stroke studies to be conducted in a more relevant 3D microenvironment is posed by ex vivo acute and organotypic brain slices. As described earlier, this model is able to largely retain the tissue structures, where multiple cell types retain most of cells’ in vivo properties and spatial organisation and intricate network organisation and function. However, next to different considerations, such as the limited culture time or maturation of acute and organotypic brain slices, respectively [36,37], the scarcity of human-derived brain slices restrict research to the used rodent-based (organotypic) brain slices, less faithfully predicting human pathophysiological mechanisms.

Second, the advent of iPSC-technology has boosted the development of 3D models of the brain, such as brain spheroids or organoids [117]. Neural organoids are self-assembled PSC-derived 3D in vitro cultures that recapitulate the developmental processes and cytoarchitecture of the developing human brain [117,118,119]. Different neural organoids and spheroids have been developed ranging from brain organoids containing multiple different brain regions, termed ‘cerebral organoids’, to brain region-specific organoids, including forebrain, midbrain, cerebellar and hippocampal and hypothalamic organoids, through the use of patterning factors [120,121,122,123,124,125,126,127,128]. Protocols to generate these brain spheroids/organoids differ in several aspects, such as the use of ECM, patterning factors or the initial cells used, which are either PSCs or neural stem/progenitor cells derived from PSCs. These differences can have implications on the complexity of the model, making them more or less suitable for certain specific applications. The use of these organoids has proven extremely useful for the study of neurodevelopment and associated pathologies, such as microcephaly, ZIKA virus infection and autism spectrum disorders [122,127,129,130,131]. Other applications include neurodegenerative disease modelling and neurotoxicity testing [124,132,133,134,135]. Though current brain spheroids and organoids are already useful tools gaining popularity in different biomedical fields, they are subject to continuous research aimed at improving their resemblance to the human brain. One of the major limitations of current organoid and spheroid models is the lack of vascularisation, causing the development of a hypoxic, necrotic core and further hampering the growth and maturation of neural organoids and spheroids [115,118,119,136,137]. Researchers are therefore trying to develop vascularised brain organoids [138,139,140] or implement microfluidic technologies (further described in section ‘microfluidics technologies’). Besides vasculature, organoids generally lack microglia [115,118,119,136,137], which have important roles in immune defence and maintenance of CNS homeostasis [141]. Recently developed differentiation protocols of iPSC-derived microglia [69,70,71,72,73] are paving the way to develop state-of-the-art immune-competent brain organoids and spheroids [142,143,144], more closely mimicking the human brain. Ischaemic stroke research would also greatly benefit from the generation of brain organoids containing vasculature (preferably with specialised BBB properties), and microglia, since it is a cerebrovascular disease with neuroinflammation being an important aspect of secondary injury after stroke. Finally, the heterogeneity of organoids, especially the cerebral organoids, in terms of size, shape and composition pose another major limitation [137,145,146]. Lower heterogeneity and enhanced reproducibility are crucial for controlled experiments and future potential screening approaches [145]. Several ways to reduce variability have been proposed, such as the use of bioreactors, avoidance of natural hydrogels (e.g., Matrigel) containing undefined factors, the use of patterning factors and starting from iPSC-derived neural stem/progenitor cells instead of iPSC to exclusively obtain cells of neuroectodermal lineage [115,127,145].

Only a few articles have been published so far, in which brain organoids or spheroids were subjected to hypoxic stimuli. To date, most studies that exposed neural organoids to low oxygen tension envisaged to study the effect of hypoxia on neurodevelopment and corticogenesis. For instance, Pasça et al. subjected brain region-specific organoids called human cortical spheroids (hCS) to hypoxia to determine the effect of oxygen deprivation on corticogenesis, to model injury in the developing brain. They found that intermediate progenitors, a specific population of cortical progenitors that are thought to contribute to the expansion of the primate cerebral cortex, were reduced following hypoxia and subsequent reoxygenation. Moreover, Kim et al. studied the effect of hypoxia on neurodevelopment [147]. They used human neural organoids, derived from neural stem cells (NSCs), and found that after hypoxia, reoxygenation was able to restore neuronal proliferation but no neuronal maturation, as shown by the retained decrease in TBR1+ cells. Similarly, Daviaud et al. subjected human cerebral organoids with dorsal forebrain specification to transient hypoxia, as a model for prenatal hypoxic injury, and demonstrated the distinct vulnerability and resilience of different neuroprogenitor subtypes [148]. They demonstrate that outer radial glia (FMA107+) and differentiating neuroblasts/immature neurons (TBR2+ and DCX+) are highly vulnerable to hypoxic injury, whereas NSCs displayed relative resilience to hypoxic injury and even provide a mechanism to replenish the stem cell pool, by shifting the cleavage plane angle favouring symmetric division. The results of the last-mentioned study were also replicated by our own studies. With the aim of developing a human neurospheroid model for ischaemic stroke, we equipped iPSC-derived neurospheroids with intrinsic bioluminescence to enable the real-time monitoring of the viability of neurospheroids subjected to OGD and were able to model OGD-mediated neurotoxicity [149]. By comparing 1-week-old with 4-week-old neurospheroids, containing a high proportion of undifferentiated NSCs and intermediate progenitors/immature neurons, respectively, it was demonstrated that 1-week-old neurospheroids were able to completely and spontaneously recover from the initial OGD-induced damage over the course of one week, unlike 4-week-old neurospheroids. These dynamics of OGD-mediated neurotoxicity of different ages of neurospheroids underscore the need for older, more mature neurospheroids for in vitro stroke research.

Furthermore, cerebral organoids have also been employed to further unravel the mechanisms underlying ischaemic injury. Iwasa et al. subjected cerebral organoids to OGD and reoxygenation and identified peroxisome proliferator-activated receptor (PPAR) signalling and pyruvate kinase isoform M2 (PKM2) as key markers of neuronal cells in response to OGD and reoxygenation [150]. In addition, Ko et al. described 3D cortical spheroids derived from primary rat cortical cells treated with OGD and reoxygenation as a model for cerebral ischaemia [151]. They demonstrated that their model successfully mimicked the ischaemic response as evidenced by the upregulated mRNA expressions of the key markers for stroke, S100B, IL-1β and MBP and additionally substantiate the role of transient cell-substrate interactions herein. Lastly, spheroid models have also been exposed to hypoxia to study the integrity of the BBB under pathological conditions. Nzou et al. made cortical spheroids with a functional BBB by mixing human primary brain endothelial cells, pericytes, astrocytes, and human iPSC-derived microglia, oligodendrocytes and neurons at a certain ratio in a hanging drop culture environment. They challenged the spheroids with a hypoxic stimulus and demonstrated that hypoxia resulted in BBB disruption, as evidenced by the altered localisation of tight and adherens junctions [106]. This further indicates the usefulness of the organoid/spheroid model in studying ischaemia in a physiologically relevant environment.

Alternative to neurospheroids and organoids, recently, scaffold-based 3D systems have also been proposed as a potential in vitro model for CNS injury, including stroke. Here, cells are embedded in a polymer-based scaffold that mimics the ECM of the brain. Lin et al. seeded SH-SY5Y cells onto a patterned gelatin scaffold and investigated the neuroprotective effects of resveratrol, an AMP-activated protein kinase (AMPK) activator, when subjected to OGD [152]. Vagaska et al. describe a model consisting of primary human NSCs dispersed in a hydrogel (i.e., Collagen-I/Matrigel) subjected to OGD or to thapsigargin, an inducer of intracellular calcium release [31]. In the same study, the difference in human NSC phenotype and damage response between 2D and 3D cultures of NSCs was assessed, suggesting that 3D models may be better predictors of the in vivo response to damage and compound cytotoxicity.

Finally, these brain spheroids/organoids and scaffold-based 3D cultures of CNS cells can take advantage of from microfluidic systems, to generate so called brain-on-a-chip models, forming the final category of existing 3D cell cultures of the brain. Brain-on-a-chip models and microfluidics technology are further discussed in the next section of ‘microfluidics technology’.

Considering the impact of dimensionality on cells’ morphology, proliferation, differentiation and electrophysiological properties under physiological conditions [31,115], it is not hard to assume that it might as well affect the behaviour of cells in response to pathological stimuli and/or therapeutic compounds. This concept was already demonstrated in the context of hepatotoxicity research, where 3D hepatocyte cultures were less susceptible to cell death when exposed to cytotoxins in comparison with 2D cultures [153]. Within the field of in vitro stroke research, important differences between 2D and 3D neural cultures are also becoming apparent. For instance, the earlier mentioned study of Vagaska et al. demonstrated the lower susceptibility to OGD-mediated damage for human NSCs grown in 3D, when compared to their 2D counterpart. The same could be concluded when thapsigargin was used as stimulus, after eliminating the possibility of reduced drug accessibility as a confounding factor [31]. In the context of the development of a 3D cortical spheroid model for cerebral ischaemia, Ko et al. confirmed that 3D cell culture models represent better normal brain models, since the neural cells in 3D maintained their healthy physiological morphology of a less activated state and suppressed mRNA expressions of pathological stroke markers S100B, IL1-β and MBP [151]. Moreover, our studies demonstrated different behavioural responses of neural cells in 2D and 3D. More specifically, the response to treatment with the pan-caspase inhibitor Z-VAD-FMK during and after OGD differed between NSCs cultured in 2D versus NSC-derived neurospheroids. Where Z-VAD-FMK conferred neuroprotection in 2D, in line with other publications, it failed to protect neurospheroids under OGD [149]. Altogether, these findings further underscore the importance of 3D models in basic as well as applied in vitro stroke research to complement conventional 2D cell cultures and in vivo animal studies.

### 3.4. Implementation of Microfluidics Technology in In Vitro Models of Ischaemic Stroke

Besides the factors described above, new technologies may also help to increase the complexity and predictive power of in vitro ischaemic stroke models (Figure 1). The newly developed ‘brain-on-a-chip’ models employ microfluidics technology to create a more physiologically relevant microenvironment for the culture of CNS cells. Through the spatial control over fluids in micro-meter sized channels, microfluidics enable (i) the co-culture of cells in a spatially controlled manner, (ii) generation of and control over (signalling) gradients and (iii) perfusion flow, contributing to an increase in physiological relevance of in vitro models [154]. These applications will be further discussed hereafter.

First, microfluidics facilitate physical separation of cellular populations and/or components on a microscale as a basis for mechanistic studies [83]. For instance, using microfluidic devices, the interaction between neuronal populations derived from different brain regions can be studied. This way, cortico-thalamic, cortico-hippocampal interactions and even interactions between three different brain regions (cortex, hippocampus and amygdala) have been established to model the brain’s complex neuronal architecture and functionality [83]. The studies using microfluidic systems to investigate brain region interactions are nicely described in the review by Nikolakopoulou et al. [83]. Besides the physical isolation of different cell populations, microfluidics are also used to separately study axons and cell bodies of neurons (Figure 2). Axons are directed to grow in microgrooves, thereby isolating axons from the cell soma. This platform allows the study of axonal biology, injury, regeneration and myelination but also synapse formation and modulation as well as viral spreading after axonal infection [155,156,157,158,159]. Specifically in the context of the stroke, a similar microfluidic set-up has been used to study the spreading neurotoxicity into undamaged brain areas [160]. Hereto, hippocampal neurons were cultured in each chamber and synaptically connected via axons traversing the microchannels. An isolated excitotoxic insult (i.e., glutamate) was delivered to neurons in one chamber, and the spreading toxicity of other synaptically connected neuronal populations could be monitored [160]. This system thus allows one to recapitulate focal ischaemia, which has been considered difficult to mimic in in vitro models.

Second, since microfluidics enable spatial control over fluids, gradients can be generated and precisely controlled [154]. This has proven particularly useful for studying angiogenesis, invasion and migration, as all are associated with molecular gradients in vivo [154]. Biochemical gradients of growth factors and cytokines also dictate differentiation patterning in vivo, making microfluidic devices suitable tools for studying early neurodevelopment [83,161,162,163]. Likewise, different microfluidic devices have been developed to establish oxygen gradients in cell and tissue cultures [164,165,166,167,168,169,170,171]. By flowing gas mixtures with desired oxygen concentrations through gas-permeable polydimethylsiloxane (PDMS) gas channels, cellular platforms, including adherent cells, brain slices and even 3D scaffold-based or spheroid models, can be rapidly and efficiently exposed to a range of oxygen concentrations as low as 0.1% O_2_ [164,165,166,167,168,169,170,171], which are of relevance for future ischaemic stroke research. Compared to a hypoxic chamber, where all cultures are exposed to the same oxygen tension, this microfluidic based system allows one to apply multiple oxygen concentrations or gradients to cultures, representing another possible approach to induce focal ischaemia by means of microfluidics technology [37,171].

Last but not least, the compartmentalisation of microfluidic devices allows the perfusion of media adjacent or through (3D) cell cultures on microfluidic chips. This perfusion ensures stable nutrient and oxygen supply and removal of waste metabolites and mimics physiological flows, such as interstitial or blood flow. Moreover, accompanying the fluid flow, physiological shear stresses are introduced, which have been demonstrated to be essential for cellular morphology and the gene expression of endothelial cells, when modelling vascularity [154]. The perfusion feature of microfluidics has also been exploited to specifically support the perfusion of brain spheroids and organoids generated on a microfluidic chip [172,173,174,175,176]. Evidently, this microfluidic platform is also ideal to recapitulate the BBB, and even the complete NVU, which is of particular interest for stroke research. The different BBB/NVU models will be described in section ‘BBB/NVU models’.

Despite the benefits of microfluidics in creating physiological relevant models and increasing the reproducibility of 3D CNS models [118], these models are nevertheless associated with several disadvantages. The fabrication of microfluidic devices typically relies on multi-step lithographic processes that are time-consuming and complex and require specialised equipment and expertise [105]. This has greatly limited the wide adoption of these systems in research. However, 3D printing might partially solve this issue by providing an alternative fabrication approach [105]. Moreover, microfluidic platforms are associated with limited scalability [105]. Currently, novel platforms are being developed, allowing the culture of multiple chips in parallel [23]. Finally, the use of microfluidics typically requires smaller amounts of media and cells compared to traditional cell culture systems. Though, cost-effective, this also poses a challenge for downstream analysis, requiring highly sensitive instruments [154].

#### BBB/NVU Models

Since ischaemic stroke is a cerebrovascular disease, the vasculature of the brain plays an essential role in the cause (i.e., obstructed blood flow by the blood clot) as well as progress of ischaemic stroke. Indeed, stroke is associated with disruption of the BBB, which under physiological conditions tightly controls the entry of molecules from the circulation into the brain, thereby ensuring homeostasis. However, as previously described, in vitro stroke models generally lack vasculature and thereby ignore this aspect of ischaemic stroke pathology. However, several models to investigate the BBB or broader, the NVU, have been developed over the years and are recently reviewed in detail by Andjelkovic [28]. Here, we will provide a brief overview of current and future BBB/NVU models, with their (potential) application in the context of stroke research.

The BBB is formed by specialised brain endothelial cells with barrier properties, surrounded by astrocytes and pericytes that support and maintain BBB function. The perivascular milieu of the BBB also includes neurons and neuronal endings and transiently present microglia/macrophages, which together with the BBB components are referred to as the NVU [28]. Depending on the availability of model systems and different applications, different BBB/NVU models have been used and developed in in vitro stroke research.

The oldest and simplest in vitro BBB model consists of a monolayer of brain endothelial cells (BECs). This model allows one to unravel specific mechanisms elicited in BECs under stroke-like conditions [89,92,177,178]. For instance, Itoh et al. used this model to determine whether BECs could be a source of free radicals after reperfusion, which are known for its detrimental effects on the brain after transient ischaemia [89]. When cultured on semi-permeable membranes, using Transwell systems (Figure 3), BEC monolayers enable the study of permeability of the BBB. Indeed, different in vitro studies examined the role of specific factors or mechanisms associated with OGD-induced barrier dysfunction using this model [90,91]. However, these represent only poor models of the BBB considering that the formation, maintenance and function of the BBB have been found to be depend on intercellular interactions with other CNS cells, with extensive body of evidence for the role of astrocyte-BEC and pericyte-BEC interactions [28,83,103,104,105,106]. Hereto, co- and tri-culture Transwell systems were developed (Figure 3), with BECs seeded on the membrane in the upper chamber, while perivascular cells (astrocytes, pericytes and possibly even neuron and microglia) are cultured either on the other side of the membrane or on the bottom of the lower chamber. Comparably to the monoculture systems, these models have been used to study OGD-related mechanisms leading to BBB alterations [179,180,181,182].

Although the co-culture Transwell systems improved BBB/NVU models to a significant extent, the lack of a 3D structure and the lack of flow and accompanying shear stress, known to be an important factor in inducing and maintaining the BBB-characteristic phenotype of BECs, limits the physiological relevance of these BBB/NVU models [28,105]. Hence, 3D models of BBB/NVU were developed, including the dynamic in vitro model of the BBB (DIV-BBB) and microfluidic BBB/NVU platforms.

The first model of the BBB/NVU able to incorporate flow was the DIV-BBB model (Figure 4). In this platform, BECs are seeded on the luminal side of artificial capillaries, i.e., microporous pronectin-coated polypropylene hollow fibres, while perivascular cells (mostly astrocytes and pericytes) were grown on the outer surface. By means of a pulsatile pump, the intraluminal flow and pressure can be obtained comparable to that found in capillaries in vivo [28,105]. This way, BECs are exposed to flow and shear stress, achieving BBB properties more similar to those in vivo than static Transwell co-culture systems. DIV-BBB has been used to mimic an ischaemic-like event in vitro, by flow cessation and reperfusion in the presence of circulating leukocytes [183,184,185]. This particular experimental set-up allowed one to assess the role of inflammation, including leukocyte activation and associated release of pro-inflammatory cytokines, in BBB failure secondary to an ischaemic-like event. Despite their broad applicability in in vitro stroke research, these models are costly and require specialised equipment, limiting their adoption in studies and their high-throughput potential [105]. In terms of physiological relevance, the thick membrane (~150 µm) of the hollow fibre wall limits direct cell–cell contact between BECs and perivascular cells and limits studies of drug transport and leukocyte transmigration [28]. To this end, microfluidic systems were introduced.

Different microfluidic-based BBB/NVU models have been developed and can be roughly categorised into 2D, 2.5D and 3D BBB/NVU models (Figure 5) [186]. The first BBB/NVU microfluidics-based model was developed by Booth et al. [187], and consists of two perpendicular-crossing channels (one luminal and one abluminal) to introduce dynamic flows, a porous (ECM-coated) membrane at the intersection of the flow channels for cell culture, and even multiple embedded electrodes to monitor the functionality of the barrier (measured by transendothelial electrical resistance or ‘TEER’). BECs and astrocytes were cultured on the luminal and abluminal sides of the porous membrane, respectively. The membranes used were much thinner than the hollow fibre walls of the DIV-BBB model, allowing improved cell–cell contact. The model of Booth et al. laid the foundation for the development of other 2D microfluidic BBB models, generally including two compartments separated by a permeable membrane, where minimum one compartment acts as a flow channel to mimic vascular blood flow [188,189,190,191]. These models can differ in terms of cell types, the presence of TEER electrodes or a peristaltic pump. To the best of our knowledge, this model has not yet been used in the context of ischaemic stroke research.

The 2.5D BBB/NVU models refer to microfluidic devices consisting of a compartment containing perivascular cells dispersed in a hydrogel matrix and another compartment containing BEC monolayers grown on ECM-coated rectangular shaped PDMS channels that are exposed to fluid flow (Figure 5) [23,192,193,194]. Micropillars create distinctions between these channels, allowing hydrogels to be confined to the brain parenchymal channel [192]. Gaps between these micropillars enable direct cell–cell contact in contrast to previously mentioned membrane-based BBB models, further improving the physiological relevance of the BBB. This model, with or without adaptations, has already been applied in the context of ischaemic stroke research in three studies, with Cho et al. being the first to suggest the use of their microfluidic BBB/NVU model as an in vitro model for ischaemic stroke [193]. They developed a BBB model, consisting of a monoculture of rat brain endothelial cell line monolayers on ECM-coated rectangular-shaped PDMS channels, without fluid flow or shear stress, and subjected it to ischaemia-like conditions by means of replacing the medium with glucose-free medium and incubation in an anaerobic chamber. They confirmed the disruption of BBB integrity under these stimuli and used this model to evaluate the protective function of antioxidant and ROCK-inhibitor treatments, which appeared to be limited [193]. Compared to this study, Lyu et al. and Wevers et al. both generated more predictive models of ischaemic stroke, by co-culturing human-based neural cells embedded in 3D hydrogels and by incorporating halted perfusion as an additional stimulus to mimic ischaemic stroke, next to hypoxia (either by OGD or chemical hypoxia) and hypoglycaemia (replacement of media by glucose-free (and serum-free) media) [23,29]. Lyu et al. developed a microphysiological model of ischaemic stroke based on a BBB/NVU model containing human BECs, pericytes, astrocytes, microglia and neurons in order to assess the neurorestorative potential of different therapeutic stem cells after ischaemic damage [29]. Wevers et al. described a human NVU on-a-chip model containing primary BECs in co-culture with iPSC-derived astrocytes and neurons, that under stroke-mimicking conditions demonstrated reduced BBB integrity, mitochondrial membrane potential and ATP, which are common features of ischaemic stroke. Moreover, they use a platform allowing the culture of 40 NVU on-a-chip models simultaneously, making the platform suitable for high-throughput applications [23].

Finally, 3D BBB/NVU microfluidic models consist of a 3D hydrogel matrix containing a cylindrical void, generated by using a needle as a mould or by means of a process called viscous fingering, that is lined with BECs on the gel’s inner surface (Figure 5) [195,196,197,198,199,200]. This allows direct cell–cell contact, without the need for micropillars or membranes. The choice of hydrogel is important, since it needs to be able to resist perfusion while providing physiologically relevant cues resembling ECM in vivo [186]. So far, this type of microfluidic BBB/NVU model has not been used yet in stroke research, but may become of significant importance in future in vitro stroke research.

## 4. Conclusions

Current in vitro models are limited by either the rodent origin, the cell line-inherent immortalised/transformed phenotype, the 2D culture method, the lack of other CNS cell types and/or the lack of perfusion flow. Throughout this review manuscript, several factors affecting the physiological relevance of in vitro models were outlined, suggesting that human-based 3D models consisting of multiple cell types may better recapitulate human ischaemic responses. The integration of different technologies, including iPSC-technology and the more recently emerging spheroid/organoid technology and advances in biomaterial research, will undoubtedly enable the further development of these models. Additionally, the implementation of microfluidics technology will allow one to mimic ischaemic stroke even more closely, e.g., by interrupted perfusion flow and/or by modelling focal ischaemia. While the relevance of these types of models are increasingly being recognised in different biomedical fields, they are now also slowly gaining momentum in the ischaemic stroke field. They have the potential to complement 2D in vitro models and animal models, each having their own strengths and limitations, to gain more insight into the pathophysiology of the ischaemic stroke. Moreover, the introduction of these models in the preclinical phase of drug discovery and development would allow one to bridge the translational gap between preclinical studies and clinical trials, increasing the chances of success for the agents to be effective in clinical setting. Nevertheless, as (engineered) human PSC-derived 3D models are a rather recent development, it remains to be demonstrated whether these models are actually better at predicting human ischaemic responses and clinical outcomes when evaluating new agents prior to their integration in the preclinical in vitro armamentarium. Therefore, a side-to-side evaluation of rodent (engineered) 3D models with their in vivo counterpart and a validation by means of ischaemic stroke patient-derived blood and cerebral spinal fluid samples and stroke imaging, may provide more insight on their translational value. Nonetheless, still with much fundamental research ahead, all evidence points toward a clear future for advanced human PSC-derived multicellular 3D models in fundamental and translational ischaemic stroke research.

## Figures and Tables

**Figure 1 ijms-23-07140-f001:**
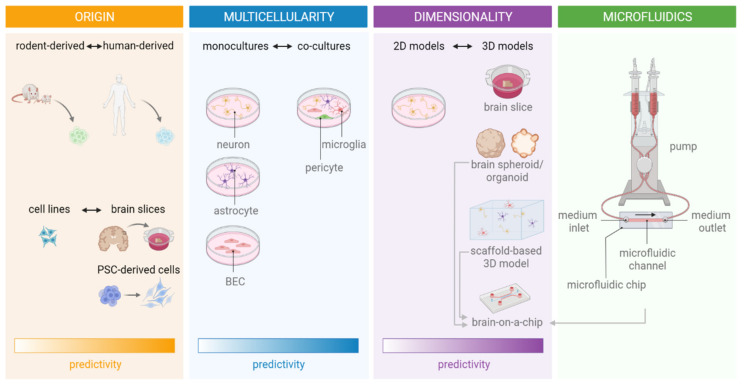
Factors defining the predictive value of in vitro ischaemic stroke models. Note that not all microfluidic devices are connected to a pump-system. (PSC, pluripotent stem cell; BEC, brain endothelial cell).

**Figure 2 ijms-23-07140-f002:**
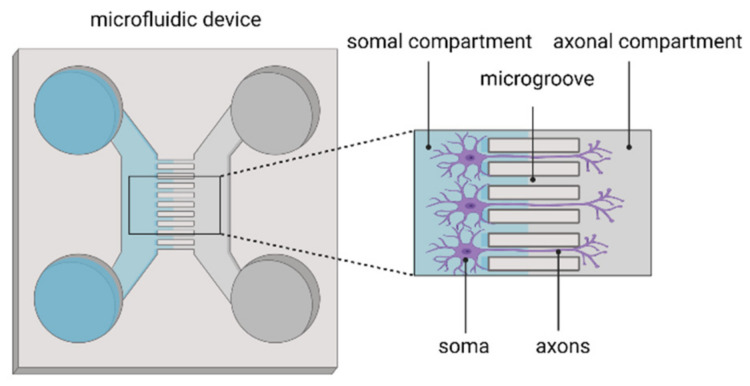
Microfluidic device for isolating axons from the neuronal soma.

**Figure 3 ijms-23-07140-f003:**
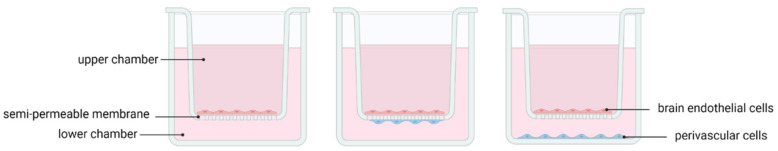
Schematic representation of BBB/NVU models employing Transwell systems. Brain endothelial cells are seeded on the semi-permeable membrane in the upper chamber. Often perivascular cells, mainly astrocytes and/or pericytes, are cultured on the other side of the membrane or on the bottom of the lower chamber.

**Figure 4 ijms-23-07140-f004:**
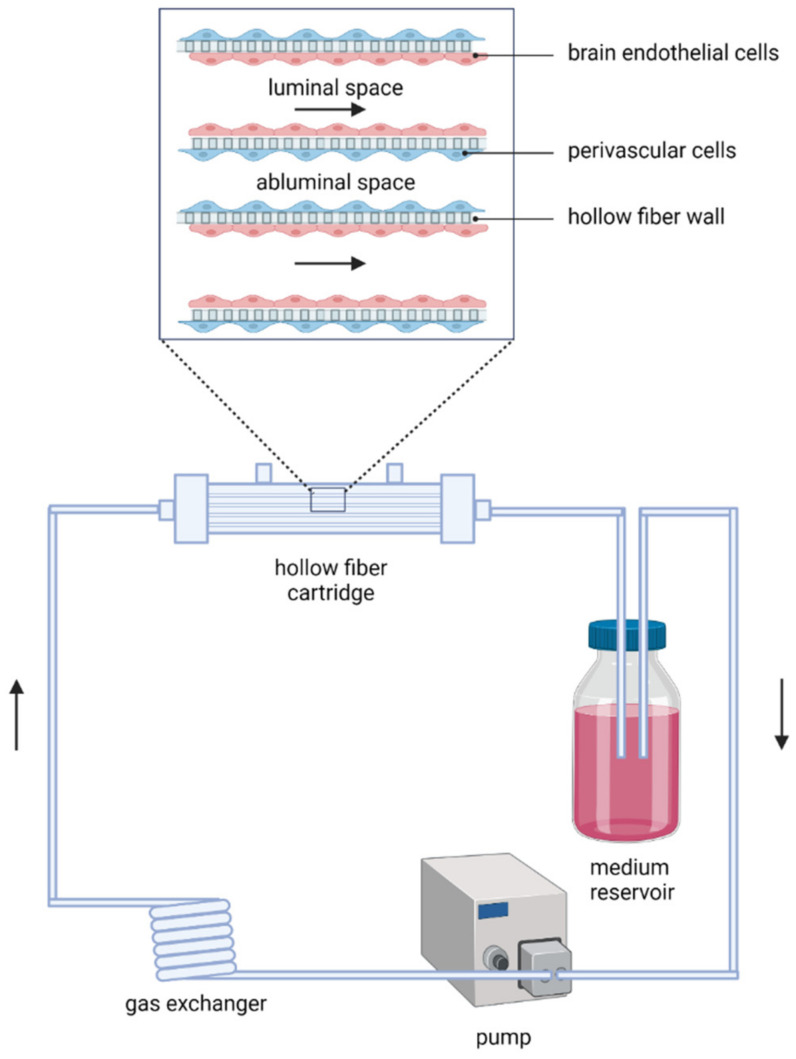
Schematic representation of the DIV-BBB model. Brain endothelial cells are seeded on the inside of ECM-coated hollow fibre structures. Perivascular cells, mainly astrocytes and/or pericytes, are cultured on the coated outer surface of the hollow fibre wall, i.e., membrane of ~150µm thick. The pulsatile pump enables the establishment of intraluminal flow and pressure comparable to that found in capillaries in vivo. (ECM, extracellular matrix).

**Figure 5 ijms-23-07140-f005:**
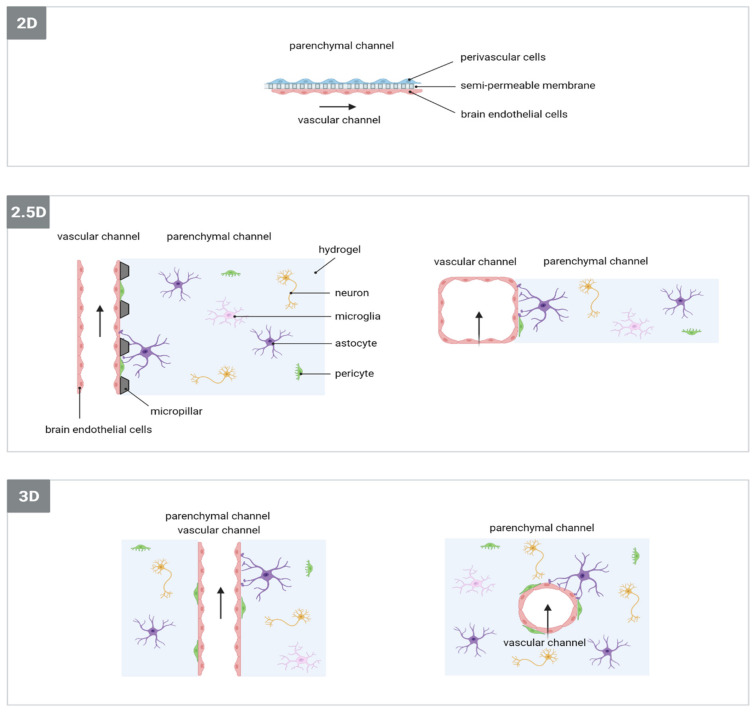
Schematic representation of 2D, 2.5D and 3D microfluidic-based BBB/NVU models. Perivascular cells indicated in 2D BBB/NVU model mainly consist of astrocytes and/or pericytes. The arrows indicate fluid flow. Figure based on Cameron et al. [186] and Katt et al. [192].

## Data Availability

Not applicable.

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
