# Peer review of "Promising Strategies for the Development of Advanced In Vitro Models with High Predictive Power in Ischaemic Stroke Research"

_ijms, 2022, doi:10.3390/ijms23137140_

Round 1
Reviewer 1 Report
The review article is well written and has covered most parts of the in vitro ischaemic stroke models.
I have only several minor comments on the manuscript as follows:
1. th abstract looks a mini introduction and needs improved. It should contain the outcome of the review, conclusions and future perspectives;
2. the authors should introduce briefly the pathology of ischaemic stroke, e.g. ischaemic core and penubra, and indicate the penumbra is the theapeutic target for ischaemic stroke, and the in vitro ischaemic stroke models mimic the condition of penumbra;
3. in the section of "multicellular co-culture models", can authors indicate what the ratio of the brain cells is? should the in vitro ischaemic stroke model have the human neuron/glia or rodent neuron/glia ratio?
4. lastly, the conclusion is a bit short, and should be expanded to have critical comments and future perspectives.
Author Response
- th abstract looks a mini introduction and needs improved. It should contain the outcome of the review, conclusions and future perspectives
We agree that this information was missing, and therefore adapted the abstract for it to contain the outcome of the review, conclusions and future perspectives, taking into account the maximum word count of 200 words. - the authors should introduce briefly the pathology of ischaemic stroke, e.g. ischaemic core and penubra, and indicate the penumbra is the theapeutic target for ischaemic stroke, and the in vitro ischaemic stroke models mimic the condition of penumbra
We briefly described the pathology of ischemic stroke (including ischemic core and penumbra) in the introduction of this revised version of the review, and mentioned that the ischemic penumbra is the therapeutic target in the introduction as well as the section 'inducing ischemia-like conditions in vitro'. In the last-mentioned section, we additionally mentioned that in vitro ischemic stroke models mimic the condition of penumbra. - in the section of "multicellular co-culture models", can authors indicate what the ratio of the brain cells is? should the in vitro ischaemic stroke model have the human neuron/glia or rodent neuron/glia ratio?
We added the ratio of glia/neurons of adult human brain in the corresponding section and mentioned that ideally this ratio should be taken into consideration in in vitro models.
4. lastly, the conclusion is a bit short, and should be expanded to have critical comments and future perspectives.
We agree that the conclusion was too short and therefore expanded the conclusion with the outcome of the review, critical comments and future perspectives.
Reviewer 2 Report
Van Breedem et al., review existing and emergent in vitro models applicable in the study of AIS. This review is comprehensive and well organized, figures are well presented. Inclusion of organoid section is especially useful.
Only minor questions:
Authors claim that the rat is the most oft used in vitro animal model used, not sure if this is accurate. Regardless, doesn't detract from the main elements of the review.
Author Response
- Authors claim that the rat is the most oft used in vitro animal model used, not sure if this is accurate. Regardless, doesn't detract from the main elements of the review.
We based this claim on a sample of 39 articles, found by using search terms ‘in vitro models’ and ‘ischemic stroke’, where 12 out of these 39 articles were using rat primary neurons in their studies. The remainder including human cell lines, mouse primary neurons, co-culture of rat primary neurons and astrocytes, etc. each represented only 1-3 studies out of these 39. Nevertheless, to rule out any doubt, we changed the sentence stating that rat is the most common used species for in vitro ischemic stroke models by using the term ‘rodent’ instead in the section 'Most common cellular platforms in in vitro stroke research'.